# Characterization and Genomic Analysis of *Fererhizobium litorale* gen. nov., sp. nov., Isolated from the Sandy Sediments of the Sea of Japan Seashore

**DOI:** 10.3390/microorganisms11102385

**Published:** 2023-09-25

**Authors:** Lyudmila Romanenko, Nadezhda Otstavnykh, Naoto Tanaka, Valeriya Kurilenko, Vasily Svetashev, Liudmila Tekutyeva, Valery Mikhailov, Marina Isaeva

**Affiliations:** 1G.B. Elyakov Pacific Institute of Bioorganic Chemistry, Far Eastern Branch, Russian Academy of Sciences, Prospect 100 Let Vladivostoku, 159, Vladivostok 690022, Russia; chernysheva.nadezhda@gmail.com (N.O.); valerie@piboc.dvo.ru (V.K.); mikhailov@piboc.dvo.ru (V.M.); 2NODAI Culture Collection Center, Tokyo University of Agriculture, 1-1-1 Sakuragaoka, Setagaya-ku, Tokyo 156-8502, Japan; n3tanaka@nodai.ac.jp; 3A.V. Zhirmunsky National Scientific Center of Marine Biology, Far Eastern Branch, Russian Academy of Sciences, Palchevskogo Street 17, Vladivostok 690041, Russia; vsvetashev@mail.ru; 4ARNIKA, Territory of PDA Nadezhdinskaya, Centralnaya St. 42, Volno-Nadezhdinskoye, Primorsky krai, Vladivostok 692481, Russia; tekuteva.la@dvfu.ru

**Keywords:** marine environment, *Rhizobiaceae*, *Fererhizobium litorale* gen. nov., sp. nov., taxonomy, sandy sediments, Japan Sea

## Abstract

The taxonomic status of two gram-negative, whitish-pigmented motile bacteria KMM 9576^T^ and KMM 9553 isolated from a sandy sediment sample from the Sea of Japan seashore was defined. Phylogenetic analysis revealed that strains KMM 9576^T^ and KMM 9553 represent a distinct lineage within the family *Rhizobiaceae,* sharing 100% 16S rRNA sequence similarity and 99.5% average nucleotide identity (ANI) to each other. The strains showed the highest 16S rRNA sequence similarities of 97.4% to *Sinorhizobium garamanticum* LMG 24692^T^, 96.9% to *Ensifer adhaerens* NBRC 100388^T^, and 96.8% to *Pararhizobium giardinii* NBRC 107135^T^. The ANI values between strain KMM 9576^T^ and *Ensifer adhaerens* NBRC 100388^T^, *Sinorhizobium fredii* USDA 205^T^, *Pararhizobium giardinii* NBRC 107135^T^, and *Rhizobium leguminosarum* NBRC 14778^T^ were 79.9%, 79.6%, 79.4%, and 79.2%, respectively. The highest core-proteome average amino acid identity (cpAAI) values of 82.1% and 83.1% were estimated between strain KMM 9576^T^ and *Rhizobium leguminosarum* NBRC 14778^T^ and ‘*Rhizobium album*’ NS-104, respectively. The DNA GC contents were calculated from a genome sequence to be 61.5% (KMM 9576^T^) and 61.4% (KMM 9553). Both strains contained the major ubiquinone Q-10 and C_18:1_*ω*7*c* as the dominant fatty acid followed by 11-methyl C_18:1_*ω*7*c* and C_19:0_ cyclo, and polar lipids consisted of phosphatidylcholine, phosphatidylethanolamine, phosphatidylglycerol, an unidentified aminophospholipid, and two unidentified phospholipids. Based on phylogenetic and phylogenomic analyses, and phenotypic characterization, strains KMM 9576^T^ and KMM 9553 are concluded to represent a novel genus and species, for which the name *Fererhizobium litorale* gen. nov., sp. nov. is proposed. The type strain of the type species is strain KMM 9576^T^ (=NRIC 0957^T^).

## 1. Introduction

The genus *Rhizobium*, the type genus of the family *Rhizobiaceae* Conn 1938 [1], first described by Frank (1889) [2] and emended by Young et al. (2001) [3], has been the object of numerous taxonomic studies for a long time. It has been reported that the 16S rRNA gene sequencing has limitations in differentiating *Rhizobiaceae* members, whereas a multilocus sequence analysis (MLSA) of housekeeping genes proves to be more effective in resolving closely related species of the taxonomic position [4,5,6,7,8]. The use of MLSA demonstrated high heterogeneity of the genera *Rhizobium* and *Agrobacterium* within the family *Rhizobiaceae*, which led to the reclassification of the species group “*Rhizobium galegae*” as a novel genus *Neorhizobium* [7]. In addition, based on the MLSA results, the genus named *Pararhizobium* was proposed to accommodate the cluster of *Rhizobium giardinii*, *R. herbae*, “*R. helanshanense,*” *R. sphaerophysae*, and *Blastobacter capsulatus* [7,8]. Recently, in the phylogenomic study based on a whole genome sequencing analysis, Kuzmanovic et al. (2022) [9] have proposed a common approach for the genera delineation in the family *Rhizobiaceae,* applying a core-genome gene phylogeny and a pairwise core-proteome average amino acid identity (cpAAI) value of approximately 86%. The application of the proposed framework resulted in the reclassification of several *Rhizobium* species, including a proposal for the new genus *Xaviernesmea* and the retrieval of *Ensifer* and *Sinorhizobium*, which had previously been unified [10,11] as separate genera [9]. Currently, the family *Rhizobiaceae* comprises 19 genera, including *Rhizobium* and related *Ensifer*, *Sinorhizobium*, *Agrobacterium*, *Pararhizobium*, *Neorhizobium*, *Shinella*, *Ciceribacter,* etc. [9]. At the time of writing, the genus *Rhizobium* contains the largest number of validly described species; 88 as listed at http://lpsn.dsmz.de/genus/rhizobia, accessed on 12 September 2023, including *Rhizobium leguminosarum* as the type species of the genus [2,9,12,13]. Most rhizobia are able to invade plants of the family *Leguminosae* and incite the production of tumors or/and root nodules where the bacteria act as intracellular nitrogen-fixing symbionts. The type species *Ensifer adhaerens* had been reported to be a predator of *Micrococcus luteus* [14]. Members of the family *Rhizobiaceae* have been mainly originated from the rhizosphere and the roots of leguminous and other agricultural crops, as well as recovered from aquatic and marine ecosystems, including deep-sea and coastal sediments as non-symbiotic microorganisms [15,16,17,18,19,20,21].

The aim of the present study was to determine the taxonomic status of two aerobic, gram-negative, motile, whitish-pigmented bacteria designated KMM 9576^T^ and KMM 9553 isolated from a sandy sediment sample obtained from the Sea of Japan seashore. Based on molecular and phenotypic data obtained, a novel genus and species, *Fererhizobium litorale* gen. nov., sp. nov. is described.

## 2. Materials and Methods

### 2.1. Isolation and Phenotypic Characterization of Bacteria

A sandy sediment sample was collected from offshore of the Sea of Japan, Russia, (42° 54.8416 N 131° 43.0430 E) at a water depth of 0.3 m on 25 October, 2012. Strains KMM 9576^T^ and KMM 9553 were isolated using the dilution plating technique on tryptic soya agar (TSA) and the seawater medium (SWM), as described in a previous paper [22]. The strains were grown aerobically on Marine Agar 2216 (MA 2216) or Marine Broth 2216 (MB 2216), Tryptic Soya Agar (TSA) or Tryptic Soya Broth (TSB), and R2A agar (all BD Difco), yeast extract-mannitol agar (YMA) or broth (YMB), containing 10 g of mannitol; 0.5 g of KH_2_PO_4_; 0.2 g of MgSO_4_ 7H_2_O; 0.1 g of NaCl; 4 g of CaCO_3_; 0.4 g of yeast extract; 15 g of agar, distilled water of 1 l, and stored at −70 °C in the MB supplemented with 20% (*v/v*) glycerol. No growth was observed when they were tested for the anaerobic growth on MA 2216 for a week using an AnaeroPack^TM^ (Mitsubishi Gas Chemical America, Inc., New York, NY, USA). Strains KMM 9553 and KMM 9576^T^ have been deposited to the Collection of Marine Microorganisms (KMM), G.B. Elyakov Pacific Institute of Bioorganic Chemistry, Far Eastern Branch of the Russian Academy of Sciences, Vladivostok, Russia, and the type strain KMM 9576^T^ was deposited to the NODAI Culture Collection Center (NRIC), Tokyo University of Agriculture, Japan, under number of NRIC 0957^T^. The type strains *Rhizobium leguminosarum* NBRC 14778^T^, *Ensifer adhaerens* NBRC 100388^T^, *Sinorhizobium garamanticum* LMG 24692^T^, *Rhizobium rhizogenes* NBRC 13257^T^, *Agrobacterium radiobacter* NBRC 13532^T^, *Pararhizobium giardinii* NBRC 107135^T^, and *Mesorhizobium mediterraneum* JCM 21565^T^ were kindly provided by the respective culture collections. All strains used in this study for phenotypic tests and lipid analyses were grown on/in YMA and YMB, if not stated otherwise. Gram-staining, oxidase, catalase, and motility (the hanging drop method) tests were examined according to the standard methods described by Smibert and Krieg (1994) [23]. The morphology of cells negatively stained with a 1% phosphotungstic acid was examined by electronic transmission microscopy Libra 120 (Carl Zeiss, Oberkochen, Germany), provided by the Far Eastern Centre of electronic microscopy, A.V. Zhirmunsky National Scientific Center of Marine Biology, Far Eastern Branch of the Russian Academy of Sciences, using cells grown in diluted (1/10) TSB on carbon-coated 200-mesh copper grids. The tests, including the hydrolysis of starch, gelatin, L-tyrosine, chitin, casein, DNA, and nitrate reduction (sulfanilic acid/α-naphthylamine test), as well as the growth at different salinities (0–10% NaCl), temperatures (5–40 °C), and pH values (4.0–11.5) were conducted as described by Smibert and Krieg (1994) [23]. The medium YMA (or YMB) was used as a basal, while mannitol and CaCO_3_ were omitted for determination of substrate hydrolysis and pH, respectively. Formation of H_2_S from thiosulfate was tested in the YMB using a lead acetate paper strip. Biochemical tests for all studied using API 20E, and strains KMM 9576^T^ and KMM 9553 using API ID32 GN (bioMérieux, Marcy-l’Étoile, France), were performed as described by the manufacturer. Carbon source utilization was performed with the API 50 CHB/E tests (bioMérieux, Marcy-l’Étoile, France) according to the manufacturer’s instructions.

Antibiotic susceptibility of strains studied was examined on YMA plates using commercial paper discs (Research Centre of Pharmacotherapy, St. Petersburg, Russia) impregnated with the following antibiotics (µg per disc, unless otherwise indicated): ampicillin (10), benzylpenicillin (10 U), vancomycin (30), gentamicin (10), kanamycin (30), carbenicillin (100), chloramphenicol (30), neomycin (30), oxacillin (10), oleandomycin (15), lincomycin (15), ofloxacin (5), rifampicin (5), polymyxin (300 U), streptomycin (30), cephazolin (30), cephalexin (30), erythromycin (15), nalidixic acid (30), tetracycline (30), and doxocycline (10). For polar lipid and fatty acid analyses, strains KMM 9576^T^ and KMM 9553 and seven related type strains were cultivated on YMA at 28 °C for 48 h. Lipids were extracted using the extraction method of Folch et al. (1957) [24]. Two-dimensional thin layer chromatography of polar lipids was conducted on Silica gel 60 F_254_ (10 × 10 cm, Merck, Darmstadt, Germany) using chloroform–methanol–water (65:25:4, *v/v*) for the first direction, and chloroform–methanol–acetic acid–water (80:12:15:4, *v/v*) for the second [25]. Lipids were detected by spraying with cerium–ammonium molybdate (CAM) containing 5 g of ammonium molybdate and 0.2 g of ceric sulphate in 100 mL of 10% sulphuric acid, followed by heating at 110 °C. Amino-containing lipids were determined with ninhydrin, phospholipids with molybdate reagent, glycolipids with alpha–naphthol, and choline-containing lipids with Dragendorff’s reagent. Respiratory lipoquinones were analyzed by the reversed-phase high-performance thin-layer chromatography as described by Mitchell and Fallon (1990) [26]. Fatty acid methyl esters (FAMEs) were prepared according to the procedure of the Microbial Identification System (MIDI) [27]. The analysis of FAMEs was performed using the GC–17A chromatograph (Shimadzu, Kyoto, Japan) equipped with a capillary column (30 m × 0.25 mm I.D.) coated with Supecowax-10 and SPB-5 phases (Supelco, Bellefonte, PA, USA). Identification of FAMEs was accomplished by equivalent chain-length values and comparing the retention times of the samples to those of standards. In addition, FAMEs were analyzed using a GLC–MS Shimadzu GC–MS model QP5050 (Column MDM–5S, the temperature program from 140 °C to 250 °C, at a rate of 2 °C/min).

### 2.2. 16S rRNA Gene Sequence and Phylogenetic Analysis

Genomic DNAs of strains KMM 9576^T^ and KMM 9553 extracted according to the method of Saito and Miura [28] were used to determine 16S rRNA gene sequences as described by Shida et al. (1997) [29]. The 16S rRNA gene sequences were compared with those of the closest relatives using EzBioCloud service [30]. Phylogenies were performed on the GGDC web server (http://ggdc.dsmz.de/, accessed on 12 September 2023) [31] using the DSMZ phylogenomics pipeline [32] adapted to single genes. Maximum likelihood (ML) and maximum parsimony (MP) trees were inferred from the alignment with RAxML [33] and TNT [34], respectively. The robustness of phylogenetic trees was estimated by the bootstrap analysis of 1000 replicates.

### 2.3. Whole-Genome Sequencing, Phylogenomic, and Comparative Analyses

The genomic DNAs were obtained from the strains KMM 9576^T^ and KMM 9553 using the High Pure PCR Template Preparation Kit (Roche, Basel, Switzerland). The quantity and quality of the genomic DNAs were measured using DNA gel electrophoresis and the Qubit 4.0 Fluorometer (Thermo Fisher Scientific, Singapore, Singapore). Preparation of the DNAs sequencing libraries was conducted using Nextera DNA Flex kits (Illumina, San Diego, CA, USA) and whole-genome sequencing were performed subsequently using paired-end runs on an Illumina MiSeq platform with a 150-bp read length. The reads were trimmed using Trimmomatic version 0.39 [35], and their quality was assessed using FastQC version 0.11.8 (https://www.bioinformatics.babraham.ac.uk/projects/fastqc/, accessed on 21 August 2021). Filtered reads were assembled into contigs with SPAdes version 3.15.3 [36], and genome metrics were calculated with the help of QUAST version 5.0.2 [37]. The genome completeness and contamination were estimated by CheckM version 1.1.3 based on the taxonomic-specific workflow (family *Rhizobiaceae*) [38].

The draft genome assemblies were annotated using NCBI Prokaryotic Genome Annotation Pipeline (PGAP) and Rapid Annotation using Subsystem Technology (RAST) [39,40]. Comparisons of the Average Nucleotide Identity (ANI), Average Amino Acid Identity (AAI), and digital DNA–DNA hybridization (dDDH) values of the strains KMM 9576^T^ and KMM 9553 and their closest neighbors were performed with the online server ANI/AAI–Matrix [41] and TYGS platform [31], respectively. Core-proteome average amino acid identity (cpAAI) values between the strains and representatives of the family *Rhizobiaceae* were computed using a pipeline proposed by Kuzmanović et al. (https://github.com/flass/cpAAI_Rhizobiaceae, accessed on 10 January 2023) [9]. The core-proteome phylogeny based on 170 non-recombining core protein markers, including various members of *Rhizobium* genus, was inferred using IQ–TREE software under the LG + F + I + I + R8 model with bootstrapping using 100 replicates [42].

Annotation of secondary metabolite biosynthetic gene clusters was conducted using antiSMASH server version 6.1.1 (https://antismash.secondarymetabolites.org/#!/start, accessed on 13 March 2023). To identify carbohydrate-active enzymes (CAZymes), the dbCAN3 meta server version 11 was used with default settings (http://cys.bios.niu.edu/dbCAN3, accessed on 27 March 2023) [43]. Predictions by two of the three algorithms integrated within the server were considered sufficient for CAZy family assignments. Genome-wide analysis of orthologous clusters was performed using OrthoVenn2 and OrthoVenn3 (https://orthovenn3.bioinfotoolkits.net/home, accessed on 31 June 2023) [44,45]. The unique genes of strains KMM 9576^T^ and KMM 9553 were functionally annotated using eggNOG-mapper v2 server (http://eggnog-mapper.embl.de/, accessed on 12 January 2023) [46].

## 3. Results and Discussion

### 3.1. Phylogenetic and Phylogenomic Analyses

The 16S rRNA gene sequences of strains KMM 9576^T^ and KMM 9553 submitted to the GenBank under accession numbers LC126306 and LC126307 were 100% identical to the 16S rRNA gene sequences ON040664 and ON040663, respectively, and retracted from their genomic sequences. Phylogenetic analysis based on 16S rRNA gene sequences revealed that strains KMM 9576^T^ and KMM 9553 are close to each other (100% identity), representing a distinct lineage within the family *Rhizobiaceae* (Figure 1). In the phylogenetic trees, *Hoeflea anabaenae* WH2K^T^ was a sister clade. However, the strains shared the highest sequence similarity to *S. garamanticum* LMG 24692^T^ (97.7%), followed by *E. adhaerens* NBRC 100388^T^ (96.9%) and *P. giardinii* NBRC 107135^T^ (96.8%). The similarity to other *Rhizobiaceae* members did not exceed 96.8%.

The ANI values between strain KMM 9576^T^ and *E. adhaerens* NBRC 100388^T^, *S. fredii* USDA 205^T^, *P. giardinii* NBRC 107135^T^, and *R. leguminosarum* NBRC 14778^T^ were 79.9%, 79.6%, 79.4%, and 79.2%, respectively. The AAI values between strains KMM 9576^T^ and KMM 9553, and members of related genera, varied from 66.7 to 69.8% (Appendix A). The AAI values fell into the range of 60–80% [47] and did not exceed a cut-off value of 75% accepted for the delineation of *Rhizobiaceae* genera [48]. Genome sequences of strains KMM 9576^T^ and KMM 9553 exhibited ANI/AAI values of the 99.5%/99.7%, and the dDDH values of 99.6%, confirming their non-clonal origin (Appendix A).

In addition, the highest cpAAI values of 82.1% and 83. 1% were estimated between strain KMM 9576^T^ and *R. leguminosarum* NBRC 14778^T^ and “*Rhizobium album*” NS-104^T^ [49], respectively, which is below a threshold value of 86% proposed by Kuzmanovic et al. (2022) [9] for the genera discrimination in the family *Rhizobiaceae* (Figure 2, Appendix A). The core-proteome phylogeny provided by the cpAAI pipeline based on 97 strains of *Rhizobiaceae* genera (github.com/flass/cpAAI_Rhizobiaceae) showed that the strains KMM 9576^T^ and KMM 9553 were more closely related to “*R. album*” NS-104^T^ and “leguminosarum–rhizogene” clades (Figure 3). The position of strain “*R. album*” NS-104 on the phylogenetic tree (Figure 3), as well as its pairwise cpAAI values (Appendix A), which indicate that it was incorrectly named and should be designated as a new *Rhizobiaceae* genus. Phylogenomic analysis confirmed that the novel strains form a distinct lineage adjacent to *R. leguminosarum* NBRC 14778^T^, which indicated that they may represent a novel genus of the family *Rhizobiaceae*.

Thus, these phylogenetic and phylogenomic data suggested that strains KMM 9576^T^ and KMM 9553 represent a novel species and a novel genus within the family *Rhizobiaceae*.

### 3.2. Genomic Characteristics

According to an evaluation by the CheckM tool [38], the genome sequences of strains KMM 9576^T^ and KMM 9553 were obtained with high completeness (98.2% and 100%) at low values of contaminations (0.04% and 0.22%), respectively. The draft genomes of KMM 9576^T^ and KMM 9553 were de novo assembled into 67 and 94 contigs, with N_50_ values of 350,002 and 211,989 bp, and L_50_ values of 5 and 8, respectively. The genome sizes were estimated to be 5,252,096 and 5,494,465 bp in length with a coverage of 61× and 41×, respectively. The genome sequences were in accordance with the proposed minimal standards for bacterial taxonomy [50]. The comparison of genome features with genomes of type strains of closely and distantly related *Rhizobiaceae* genera is presented in Table 1. The genome sequences contain from 4981 (KMM 9576^T^) to 7623 (*R*. *leguminosarum* USDA 2370^T^) genes, from 45 (KMM 9576^T^) to 64 (*E. adhaerens* ATCC 33212^T^) tRNAs, and from 1 up to 5 ribosomal RNA operon copies (*E. adhaerens* ATCC 33212^T^).

According to the RAST annotation, about 1400 genes for both strains KMM 9576^T^ and KMM 9553 are in subsystems, among which the largest number of genes was assigned to “Amino Acids and Derivatives,” “Carbohydrates,” and “Cofactors, Vitamins, Prosthetic Groups, Pigments.” According to the KEGG annotation, both strains have genes encoding the Embden–Meyerhof pathway (except for phosphofructokinase and fructose–bisphosphatase ones, but diphosphate-dependent phosphofructokinase is present), the pentose phosphate pathway, the Krebs cycle, and the glyoxylate cycle. In addition, there is a complete pathway for glycogen biosynthesis and degradation. According to the dbCAN3 server, KMM 9576^T^ and KMM 9553 encode a total of 109 and 113 CAZymes, respectively, with glycosyltransferases (GT) being more abundant in the GT2 and GT14 families.

To identify *Fererhizobium* specific genes, orthologous groups were analyzed using the translated proteomes of KMM 9576^T^ and KMM 9553 in comparison with those of the three closely related genera “*R. album*”, *R. leguminosarum*, and *R. rhizogenes* (Figure 3). A total of 6487 orthologous clusters comprising 31,339 proteins were found. The core genome shared by the five compared strains was represented by 2547 orthologous clusters (Figure 4a). Among them, 1781 were single-copy gene clusters.

The distinctive feature of the *Fererhizobium* core genome was an excessive number of genes in the clusters of L-arabinose transport ATP-binding protein (GO:0015407; 11–12 genes versus 1–3), and putative adenylate cyclase 3 (GO: 0035556; 9–10 genes versus one). In pairwise comparison, the largest number of orthologous clusters of 963 was revealed for the pair KMM 9576^T^/KMM 9553, following *R. rhizogenes*/*R. leguminosarum* (694), “*R. album*”/*R. leguminosarum* (248), and “*R. album*”/*R. rhizogenes* (145). Both *Fererhizobium* strains shared the most of clusters with “*R. album*” NS-104^T^ (246), followed by *R. leguminosarum* NBRC 13257^T^ (203) and *R. rhizogenes* NBRC 13257^T^ (87) (Figure 4a). Among 963 *Fererhizobium*-specific orthologous clusters, functions of the most were associated with biological processes, such as transcription (GO:0006351) and regulation of transcription (GO:0006355), transmembrane transport (GO:0055085), and sodium ion transport (GO:0006814). A number of specific clusters (Figure 4b) for every strain were 128 for “*R. album*” NS-104^T^, 93 for *R. leguminosarum* NBRC 14778^T^, 70 for *R. rhizogenes* NBRC 13257^T^, and only 7 for KMM 9553. According to the RAST and eggNOG-mapper annotations, more than 50% of singletons were not annotated. Singletons for both strains KMM 9576^T^ and KMM 9553 were mostly enriched in genes encoding mobile or selfish elements, restriction–modification systems, transport systems, and regulatory elements. Both strains do not have key nitrogen-fixation (nifHDK) and nodulation (nodABC) genes in their genomes. In addition, genes that are involved in the biosynthesis of capsular polysaccharide, exopolysaccharide, and lipopolysaccharide were revealed in KMM 9576^T^ and KMM 9553. According to antiSMASH, the annotation of secondary metabolite biosynthetic gene clusters uncovered two homoserine lactone biosynthetic clusters, one redox-cofactor, such as PQQ (NC_021985:1458906-1494876), one thioamitide RiPP, as found in JOBF01000011, one other unspecified ribosomally synthesized and post-translationally modified peptide product (RiPP), one N-acyl amino acid, and one terpene biosynthetic cluster.

### 3.3. Phenotypic Characterization and Chemotaxonomy

Bacteria KMM 9576^T^ and KMM 9553 were small rod- or ovoid-shaped motile by one or two subpolar flagella, and cells containing more than two flagella were also observed (Figure 5a,c). Novel bacteria could grow in salinities from 0 to 4% and grew well on/in SWM, MA 2216, MB 2216, R2A agar, TSB, TSA, YMA, and YMB. On carbohydrate-containing media, novel strains produced copious extracellular material, particularly strain KMM 9553. Individual cells of strain KMM 9553 could be observed as non-motile probably due to the large amount of mucus the strain tends to produce (Figure 5b).

The novel strains were not identical in their phenotypic traits. Strain KMM 9576^T^, unlike KMM 9553, demonstrated a weak reaction of DNA hydrolysis, which tested negative for the oxidation of L-arabinose in API 20E and proved to be resistant to cephazolin and oleandomycin (Table 2) and utilized different carbon sources in API 50 CHB/E tests (Appendix A). Strains KMM 9576^T^ and KMM 9553 were similar to *A. radiobacter* NBRC 13532^T^ in being able to grow in 0–4% of NaCl, and to hydrolyze DNA. Unlike other relatives, strains KMM 9576^T^ and KMM 9553 and *R. leguminosarum* NBRC 14778^T^ were able to produce H_2_S and hydrolyze tyrosine (Table 2). Strains KMM 9576^T^ and KMM 9553 differed from *E. adhaerens* NBRC 100388^T^ and *P. giardinii* NBRC 107135^T^ in not being able to grow in 2–4% of NaCl and in their carbohydrate utilization patterns (Table 2 and Appendix A).

Both strains KMM 9576^T^ and KMM 9553 contained Q-10 as the major ubiquinone; the predominant fatty acid was detected to be C_18:1_*ω*7*c* (52.47 and 46.16%), followed by 11-Methyl C_18:1_*ω*7*c* (14.91 and 14.11%), C_19:0_cyclo (13.01 and 15.27%), C_18:0_ (10.98 and 7.09%), and C_14:0_ 3-OH (3.86 and 12.77%) in both strains, respectively (Table 3). The fatty acid profiles of strains KMM 9576^T^ and KMM 9553 were found to be similar to that of *R. leguminosarum* NBRC 14778^T^. Strains *E. adhaerens,* NBRC 100388^T^, and *S. garamanticum* LMG 24692^T^ differed from the novel strains in the presence of small amounts of C_16:0_ 3-OH, C_18:0_ 3-OH, and C_17:0_ cyclo (in *E. adhaerens* NBRC 100388^T^), while *P. giardinii* NBRC 107135^T^ differed in the presence of C_18:0_ 3-OH and C_17:0_ cyclo, as shown in Table 3. Strains *M. mediterraneum* JCM 21565^T^ did not contain C_14:0_ 3-OH and revealed a distinction in the presence of iso-C_17:0_ as compared with other related strains (Table 3). *R. rhizogenes* NBRC 13257^T^ was significantly distinguished from all bacteria studied by the content of C_15:0_ 3-OH and C_19:0_. These findings corroborate the results reported by Tighe et al. [51]. Unlike related type strains, *R. rhizogenes* NBRC 13257^T^ and *A. radiobacter* NBRC 13532^T^ contained high proportions of C_14:0_ 3-OH and C_16:0_ 3-OH, whereas novel strains contained a high amount of 11-Methyl C_18:1_*ω*7*c* (Table 3).

The polar lipid compositions of strains KMM 9576^T^, KMM 9553, and related bacteria tested did not significantly differ and included phosphatidylcholine (PC), phosphatidylglycerol (PG), phosphatidylethanolamine (PE), an unidentified aminophospholipid (APL), and two unidentified phospholipids (PL and PL1) (Appendix A). The type strains *R. rhizogenes* NBRC 13257^T^, *S. garamanticum* LMG 24692^T^, *E. adhaerens* NBRC 100388^T^, and *A. radiobacter* NBRC 13532^T^ differed from novel strains by the presence of additional unidentified phospholipids (PL2) in their lipid compositions. *R. rhizogenes* NBRC 13257^T^ and *S. garamanticum* LMG 24692^T^ contained additional unidentified lipids (L and/or L1). *M. mediterraneum* JCM 21565^T^ contained a minor amount of PE compared to those of novel strains and included an unidentified lipid (L2) (Appendix A).

The DNA GC contents of 61.4% and 61.5% were calculated from a genome sequence of strains KMM 9576^T^ and KMM 9553, respectively, which is close to the values of 49–68 mol% reported for the members of the family *Rhizobiaceae* [1]. The phylogenetic distinctness of strains KMM 9576^T^ and KMM 9553 was supported by phenotypic differences in the temperature and salinity ranges that provided their growth, substrate hydrolysis ability, and carbohydrate utilization patterns. Differential phenotypic and physiological characteristics are indicated in Table 2 and Appendix A. Based on the combination of phylogenomic analyses and phenotypic characteristics, it is proposed to classify strains KMM 9576^T^ and KMM 9553 as a novel genus and species, *Fererhizobium litorale*.

## 4. Conclusions


**Description of *Fererhizobium* gen. nov.**


*Fererhizobium* gen. nov. (Fe.re.rhi.zo’bi.um. L. adv. *fere*, nearly, almost; N.L. neut. n. *Rhizobium* a bacterial generic name; N.L. neut. n. *Fererhizobium*, a genus adjacent to *Rhizobium*).

Gram-stain negative, aerobic non-spore-forming rod-shaped or ovoid cells. Oxidase- and catalase-positive. The major respiratory quinone is ubiquinone-10. The major fatty acid is C_18:1_*ω*7*c*, followed by 11-Methyl C_18:1_*ω*7*c* and C_19:0_cyclo. The polar lipids include phosphatidylcholine, phosphatidylethanolamine, phosphatidylglycerol, an unidentified aminophospholipid, and two unidentified phospholipids. Phylogenetically related to members of the genera *Rhizobium*, *Sinorhizobium*, and *Pararhizobium* within the family *Rhizobiaceae*, class *Alphaproteobacteria.* The type species is *Fererhizobium litorale* sp. nov.


**Description of *Fererhizobium litorale* sp. nov.**


*Fererhizobium litorale* (li.to.ra’le. L. neut. adj. *litorale* of or belonging to the seashore).

In addition to the properties described for the genus, the species is characterized as follows. Whitish-pigmented, rod- or ovoid-shaped cells, 0.5–0.7 μm in diameter and 0.8–1.4 μm in length, encapsulated, motile by one or two subpolar flagella. Cells may be observed as non-motile, with growth on/in YMA, YMB, TSA, TSB, MA 2216, MB 2216, and SWM. On carbohydrate-containing media, extracellular material is produced abundantly. Growth occurs in 0–4% of NaCl (optimal is 0.5–1%), and at 5–38 °C (optimal is 28–30 °C); there is weak growth at 38 °C and no growth at 40 °C. The pH range for growth is 6.0–10.0, with an optimum of 7.0–8.0. It is negative for hydrolysis of gelatin, casein, starch, chitin, Tween 80, and nitrate reduction, and positive for DNA hydrolysis and H_2_S production in conventional tests. On the L-tyrosine-containing medium, it forms a transparent zone but does not produce black–brown pigments.

According to the ID32 GN, it is positive for the assimilation of L-rhamnose, N-acetylglucosamine, D-sucrose, inositol, D-maltose, lactic acid, L-alanine, D-mannitol, D-glucose, salicin, L-fucose, D-sorbitol, L-arabinose, propionic acid, L-histidine, potassium 2-ketogluconate, and L-proline; weakly positive for the assimilation of D-ribose and 3-hydroxybutyric acid; and negative for the assimilation of itaconic acid, suberic acid, sodium malonate, sodium acetate, potassium 5-ketogluconate, glycogen, 3-hydroxybenzoic acid, L-serine, D-melibiose, capric acid, valeric acid, trisodium citrate, and 4-hydroxybutyric acid.

According to the API 20E, tests were positive for citrate utilization, PNPG test, urease production under anaerobic conditions, and oxidation of D-sucrose (weak reaction), while they were negative for gelatin hydrolysis, arginine dihydrolase, lysine decarboxylase, ornithine decarboxylase, H_2_S production under anaerobic conditions and tryptophane deaminase, indole production, acetoin production (Voges–Proskauer reaction), and an oxidation of D-glucose, D-mannitol, inositol, D-sorbitol, L-rhamnose, D-melibiose, and amygdalin; oxidation of L-arabinose is strain-dependent (reaction of the type strain is negative).

According to the API 50CH/E tests, it was positive for the utilization of aesculin ferric citrate, D-fucose, D-arabinose, D-ribose, and L-rhamnose; negative for the utilization of glycerol, erythritol, D-xylose, D-adonitol, L-xylose, methyl-βD-xylopyranoside, D-galactose, D-glucose, D-fructose, L-sorbose, dulcitol, inositol, D-mannitol, methyl-αD-mannopyranoside, methyl-αD-glucopyranoside, N-acetylglucosamine, amygdalin, arbutin, salicin, D-cellobiose, D-maltose, D-lactose, D-melibiose, D-sucrose, D-tregalose, inulin, D-melezitose, D-raffinose, amidon, glycogen, xylitol, gentiobiose, D-turanose, D-lyxose, D-tagatose, D-arabitol, L-arabitol, potassium gluconate, potassium 2-ketogluconate, and potassium 5-ketogluconate.

The utilization of L-arabinose, D-sorbitol, D-mannose, and L-fucose is strain-dependent (negative reaction of the type strain for L-arabinose, D-sorbitol, D-mannose utilization, and position for L-fucose utilization). Susceptible to (content per disc): gentamicin (10 µg), kanamycin (30 µg), neomycin (30 µg), ofloxacin (5 µg), rifampicin (5 µg), streptomycin (30 µg), and cephalexin (30 µg); and resistant to ampicillin (10 µg), benzylpenicillin (10 U), vancomycin (30 µg), carbenicillin (100 µg), lincomycin (15 µg), chloramphenicol (30 µg), nalidixic acid (30 µg), oxacillin (10 µg), polymyxin B (300 U), tetracycline (30 µg), doxocycline (10 µg), and erythromycin (15 µg); susceptibility to cephazolin (30 µg) and oleandomycin (15 µg) is strain-dependent (the type strain is resistant). The DNA GC content of 61.4–61.5% is calculated from the genome sequence. The major isoprenoid quinone is ubiquinone Q-10. The major fatty acid is C_18:1_*ω*7*c*, followed by 11-Methyl C_18:1_*ω*7*c* and C_19:0_cyclo. The polar lipids included phosphatidylcholine, phosphatidylethanolamine, phosphatidylglycerol, an unidentified aminophospholipid, and two unidentified phospholipids.

The GenBank accession number for the whole-genome shotgun sequence of strain KMM 9576^T^ is JALDYZ010000000.

## Figures and Tables

**Figure 1 microorganisms-11-02385-f001:**
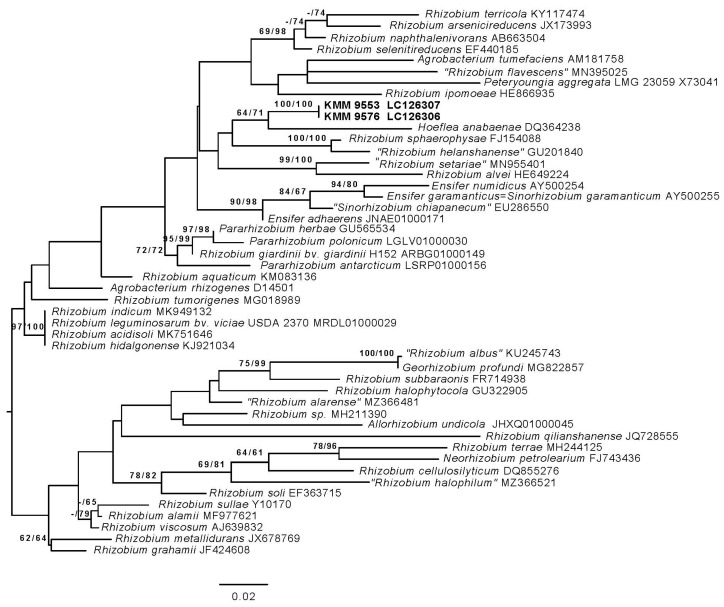
ML tree based on 16S rRNA gene sequences available from the GenBank database showing relationships of the new strains KMM 9576^T^, KMM 9553 (in bold), and related members of the family *Rhizobiaceae*. The tree was inferred under the GTR + GAMMA model and rooted by midpoint rooting. The branches are scaled in terms of the expected number of substitutions per site. The numbers above the branches are support values when larger than 60% from ML (left) and MP (right) bootstrapping (1000 replicates).

**Figure 2 microorganisms-11-02385-f002:**
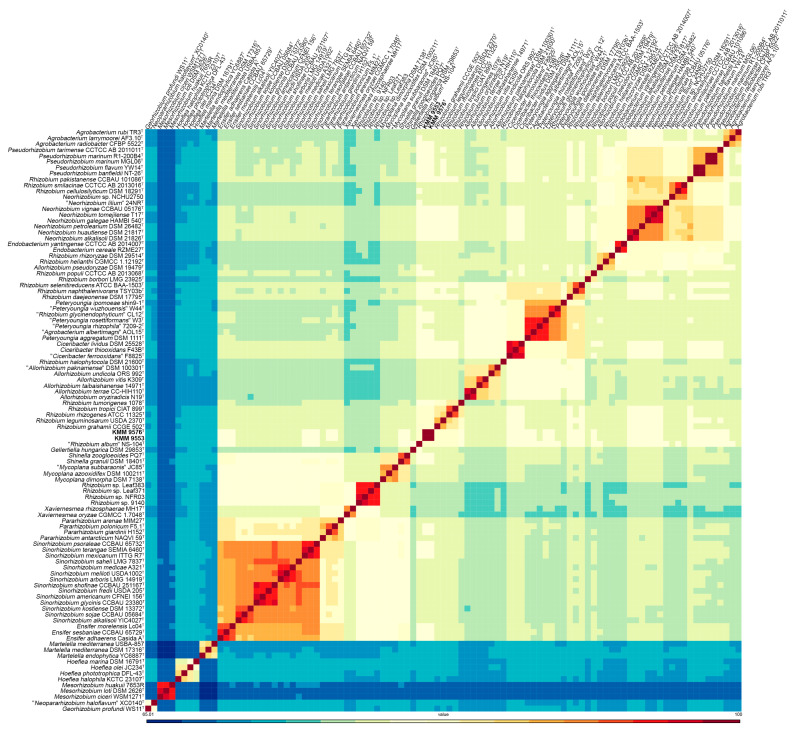
Heatmap of pairwise cpAAI values between the strains KMM 9576^T^ and KMM 9553 and the members of *Rhizobiaceae* clades. The *Mesorhizobium* spp. strains and *Georhizobium profindi* WS11 were included as outgroups. The KMM 9576^T^ and KMM 9553 are marked in bold.

**Figure 3 microorganisms-11-02385-f003:**
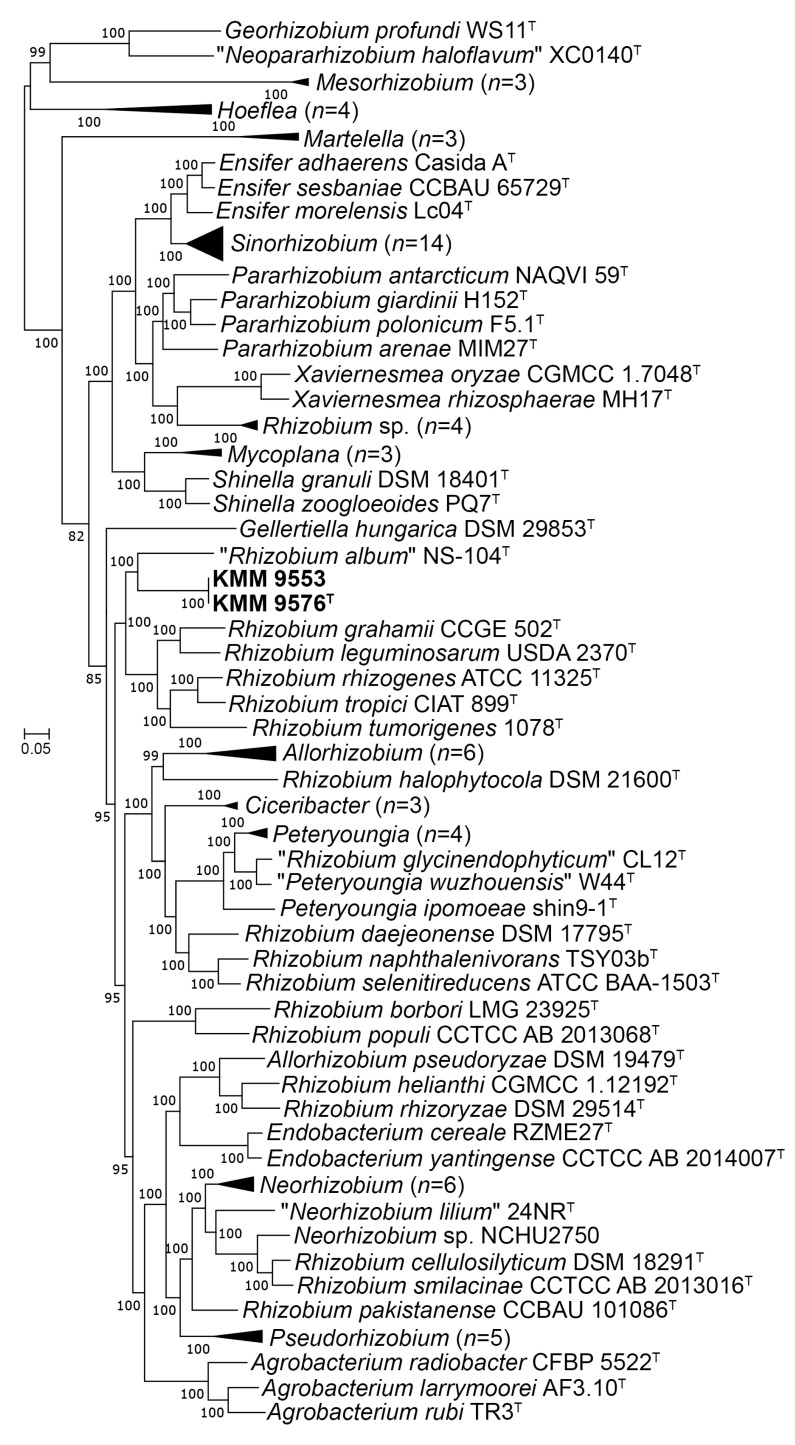
ML tree based on concatenated 170 protein sequences showing a phylogenetic position of the novel strains KMM 9576^T^, KMM 9553, and related members of the family *Rhizobiaceae*. The tree was inferred using IQ–TREE software under the LG + F + I + I + R8 model with bootstrapping of 100 replicates. The *Mesorhizobium* spp. strains and *Georhizobium profindi* WS11 were used as outgroups. Bar, 0.05 substitutions per amino acid position. Collapsed branches are shown in black triangles.

**Figure 4 microorganisms-11-02385-f004:**
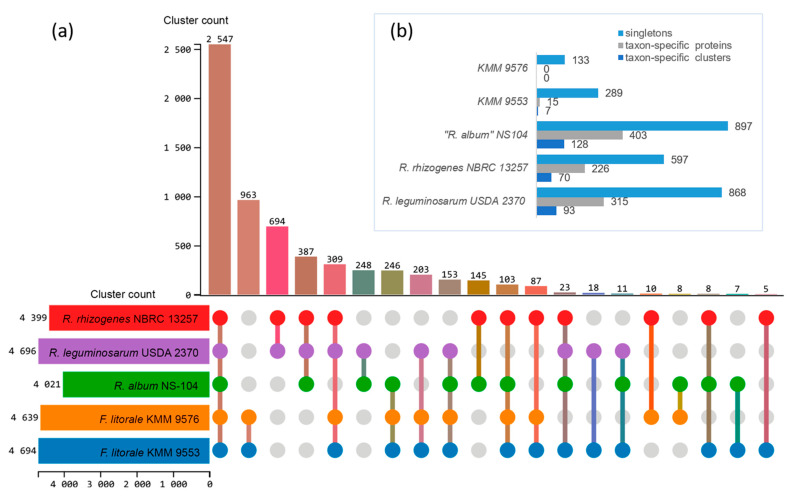
Proteome comparison of *Fererhizobium* strains and type strains of closely related genera “*R. album*” NS-104^T^, *R. leguminosarum* NBRC 14778^T^, and *R. rhizogenes* NBRC 13257^T^ using OrthoVenn3. UpSet–JS table displays common (**a**) and unique (**b**) orthologous clusters, and singletons (**b**). Horizontal bar chart on the left shows the number of orthologous clusters per strain. Right vertical bar chart indicates the number of orthologous clusters shared among the strains. The lines represent intersecting sets. Upper horizontal bar chart (**b**) shows the number of taxon-specific clusters including their proteins and singletons per strain.

**Figure 5 microorganisms-11-02385-f005:**
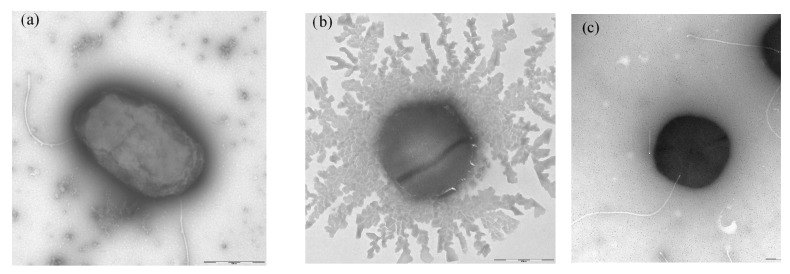
Transmission electron micrographs of strains (**a**) KMM 9576^T^ and (**b**,**c**) KMM 9553. Bar, 500 nm.

**Table 1 microorganisms-11-02385-t001:** Genomic features of strains KMM 9576^T^, KMM 9553, and type strains of closely related *Rhizobiaceae* genera.

Feature	1	2	3	4	5	6	7	8	9
Assembly level	contig	contig	contig	scaffolds	scaffolds	scaffolds	contig	contig	contig
Genome size (Mb)	5.3	5.5	7.9	7.3	6.8	6.8	5.5	7.0	6.9
Number of contigs	67	94	108	3	3	190	27	53	62
G + C Content (mol%)	61.5	61.4	60.5	62	61	60.5	59	59.5	61.5
N50 (Kb)	350	212	413	4071	4218	312	481	476	303
L50	5	8	5	1	1	8	4	7	7
Coverage	61×	41×	50×	75×	592×	-	54×	92×	12×
Total genes	4981	5244	7623	6937	6424	6631	5396	6675	6685
Protein coding genes	4819	5054	7246	6722	6111	6275	5146	6621	6438
rRNAs (5S/16S/23S)	1/1/1	1/1/1	1/1/1	5/5/5	3/3/3	2	1/1/1	1/1/1	1/1/1
tRNA	45	46	46	64	55	48	51	51	49
checkM completeness (%)	98.24	100	99.69	94.94	96.29	99.28	94.71	99.78	97.29
checkM contamination (%)	0.04	0.22	0.55	4.5	1.61	1.37	6.06	1.32	2.41
WGS project	JALDYZ01	JALDYY01	QBLB01	JNAE01	CP120373, CP120374, CP120375	ARBG01	VZPW01	BAYX01	QFBC01
Genome assembly	ASM3002890 v1	ASM3002885 v1	ASM305838 v1	ASM69796 v2	ASM2989206 v1	ASM37960 v1	ASM880138 v1	ASM69609 v1	ASM312232 v1

Strains: **1**, KMM 9576^T^; **2**, KMM 9553; **3**, *R. leguminosarum* USDA 2370^T^; **4**, *E. adhaerens* ATCC 33212^T^ (Casida A); **5**, *E. garamanticus* LMG 24692^T^ (=*S. garamanticum* LMG 24692^T^); **6**, *P. giardinii* H152^T^ (*R. giardinii* bv. *giardinii* H152^T^); **7**, *A. radiobacter* CCUG 3354^T^ (=*A. tumefaciens* CCUG 3354^T^); **8**, *R. rhizogenes* NBRC 13257^T^ (=*A. rhizogenes* NBRC 13257^T^); **9**, “*R. album*” NS-104^T^.

**Table 2 microorganisms-11-02385-t002:** Differential characteristics of strains KMM 9576^T^ and KMM 9553 and type strains of related genera.

Feature	1	2	3	4	5	6	7	8	9
Growth at/in:									
37 °C	+	+	+	+	+	+	+	+	-
38 °C	(+)	(+)	(+)	+	+	+	+	+	-
40 °C	-	-	-	(+)	-	(+)	+	+	-
2% NaCl	+	+	-	+	+	+	+	-	-
3% NaCl	+	+	-	(+)	-	-	+	-	-
4% NaCl	+	+	-	-	-	-	+	-	-
Growth on TSA	+	+	-	+	+	-	+	+	-
Tyrosine hydrolysis	+	+	+	(+)	+	-	-	-	-
DNA hydrolysis	(+)	+	-	-	-	-	+	+	-
Simmon’s citrate test	-	-	+	-	-	+	-	+	-
H2S production	+	+	+	-	-	-	-	-	-
Nitrate reduction	-	-	-	-	-	-	+	+	-
API 20E tests:									
ONPG	+	+	+	+	+	(+)	(+)	-	-
Citrate	+	+	+	-	-	-	+	+	-
Urease	+	+	+	+	-	-	+	+	-
Glucose	-	-	-	+	-	-	-	+	-
Mannitol	-	-	-	-	-	-	+	-	-
Inositol	-	-	-	+	-	-	+	-	-
Sorbitol	-	-	-	(+)	-	-	+	-	-
L-rhamnose	-	-	+	+	-	+	+	+	-
D-sucrose	(+)	(+)	-	-	-	-	+	-	-
D-melibiose	-	-	-	(+)	-	+	+	+	-
Amygdalin	-	-	-	+	-	+	+	-	-
L-arabinose	-	+	-	-	-	+	+	+	-
Sensitivity to antibiotics:									
Ampicillin	-	-	+	-	-	-	-	+	-
Vancomycin	+	+	+	+	+	+	+	+	+
Chloramphenicol	-	-	+	-	-	-	-	+	-
Kanamycin	+	+	+	-	-	+	+	+	+
Carbenicillin	+	(+)	+	-	-	+	+	+	-
Neomycin	+	+	+	+	+	+	+	+	+
Tetracycline	+	+	+	+	+	+	+	+	+
Oleandomycin	-	+	-	-	+	-	-	+	-
Oxacillin	+	(+)	+	-	-	-	-	+	-
Streptomycin	+	+	+	-	+	+	-	-	+
Cephazolin	-	+	-	-	-	-	-	+	-
Cephalexin	+	+	+	-	-	+	-	+	-

Strains: **1**, KMM 9576^T^; **2**, KMM 9553; **3**, *R. leguminosarum* NBRC 14778^T^; **4**, *E. adhaerens* NBRC 100388^T^; **5**, *S. garamanticum* LMG 24692^T^; **6**, *P. giardinii* NBRC 107135^T^; **7**, *A. radiobacter* NBRC 13532^T^; **8**, *R. rhizogenes* NBRC 13257^T^; **9**, *M. mediterraneum* JCM 21565^T^ (data were obtained from the present study). Symbol: (+) weak reaction.

**Table 3 microorganisms-11-02385-t003:** Cellular fatty acid composition (%) of strains KMM 9576^T^ and KMM 9553 and type strains of related *Rhizobiaceae* members.

Fatty Acid	1	2	3	4	5	6	7	8	9
C_14:0_	0.16	0.31	-	-	0.18	-	1.07	-	0.34
C_14:0_ 3-OH	12.77	3.86	2.51	7.84	19.50	12.39	20.95	19.50	-
C_16:1_*ω*7*c*	0.18	-	0.25	0.43	1.19	2.30	-	-	0.68
C_16:0_	3.79	4.10	8.29	13.11	8.25	17.23	16.58	12.53	13.13
C_15:0_ 3-OH	-	-	-	-	-	-	-	15.91	-
iso-C_17:0_	-	-	-	-	-	-	-	-	3.41
C_17:0_ cyclo	-	-	-	1.41	-	2.97	3.80	-	0.82
C_16:0_ 3-OH	-	-	-	1.67	-	0.20	11.63	17.14	-
C_18:1_*ω*9*c*	0.48	0.37	0.27	1.12	3.28	1.02	1.02	-	1.67
C_18:1_*ω*7*c*	46.16	52.47	61.81	51.35	42.17	47.38	17.32	6.24	34.94
C_18:0_	7.09	10.98	5.90	2.96	3.29	3.22	0.87	0.59	3.57
11-Methyl C_18:1_*ω*7*c*	14.11	14.91	5.01	8.48	7.03	5.58	0.30	-	13.46
C_19:0_ cyclo	15.27	13.01	15.96	10.47	9.75	5.94	26.62	20.98	27.99
C_18:0_ 3-OH	-	-	-	1.14	4.25	1.77	-	0.90	-
C_19:0_	-	-	-	-	-	-	-	6.81	-

Strains: **1**, KMM 9576^T^; **2**, KMM 9553; **3**, *R. leguminosarum* NBRC 14778^T^; **4**, *E. adhaerens* NBRC 100388^T^; **5**, *S. garamanticum* LMG 24692^T^; **6**, *P. giardinii* NBRC 107135^T^; **7**, *A. radiobacter* NBRC 13532^T^; **8**, *R. rhizogenes* NBRC 13257^T^; **9**, *M. mediterraneum* JCM 21565^T^ (all results were obtained from the present study). -, Not detected. Fatty acids representing <1% in all strains tested are not shown.

## Data Availability

The type strain of the species is strain KMM 9576^T^ (=NRIC 0957^T^), which was isolated from a sandy sediment sampled from the Sea of Japan seashore, Russia. The DDBJ/ENA/GenBank accession numbers for the 16S rRNA gene and the whole-genome shotgun sequences of strains KMM 9576^T^ and KMM 9553 are LC126306 (ON040664) and LC126307 (ON040663), as well as JALDYZ010000000 and JALDYY010000000, respectively.

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
