# Peer review of "Characterization and Genomic Analysis of *Fererhizobium litorale* gen. nov., sp. nov., Isolated from the Sandy Sediments of the Sea of Japan Seashore"

_microorganisms, 2023, doi:10.3390/microorganisms11102385_

Round 1

Reviewer 1 Report

 In this study, the authors present a compelling exploration of two newly isolated bacterial strains, offering a proposition for the establishment of a novel genus and species, specifically termed Fererhizobium litorale gen. nov., sp. nov. The paper delves comprehensively into the genomic attributes of these strains, comparing them with type strains from closely related genera. The overall quality of the manuscript is commendable, hinging on robust and well-defined outcomes. However, I would like to highlight a few minor points for consideration, which I outline below:

 The authors skillfully argue for the introduction of the novel strains as a distinct genus based on the cpAAI criterion. This approach is in alignment with prior research indicating the limitations of 16S rRNA sequence-based phylogenetic analyses in precisely elucidating the lineage within a specific clade. Nevertheless, I suggest that for a clearer understanding of the relationship between the new strains and closely related species, a reanalysis of the 16S rRNA gene sequence-based phylogenetic tree (Figure 1) or a phylogenomic tree (Figure 2) is advisable. This reanalysis should include supplementary data containing a sufficient number of sequences from validly published Rhizobium species. Such an addition would substantially enhance the comprehension of the phylogenetic framework.

 Regarding Figure 1, it is recommned to consider alternate methods like Maximum Likelihood and Bayesian approaches, which incorporate sequence evolution models and are better suited for constructing phylogenies. Please show Maximum Likelihood tree and others in supplementary materials. Considering that the Molecular Evolutionary Genetics Analysis software (MEGA) has been updated to version 11.0.13, it is recommended to employ the latest version along with its corresponding references.

 A minor issue arises with the resolution of Figure 3, which currently stands at a level that impedes smooth interpretation.

An observation regarding line 271: "The core-genome shared by the four compared strains was represented by 2547 orthologous clusters (Figure 4a)." Might be 5 compared strains?

 It would greatly enhance clarity if the symbols denoted by '(+)' within the parentheses in Table 2 were explicitly explained. At present, their meanings remain unspecified in the table legend.

 I recommend providing details about the cultivation conditions, as they could impact biosynthesis and subsequently alter chemotaxonomic characteristics like fatty acid profiles in bacteria. To ensure comparability, it is important to specify whether the experiments were conducted under the same conditions as the comparison strains. In this regard, please describe the incubation conditions and duration of cell incubation for the chemotaxonomic tests.

 Regarding line 381 and the term 'aerobic', I would appreciate clarification on whether the authors explored anaerobic growth conditions. It might be more accurate to denote 'strictly aerobic' or 'facultative anaerobic' growth conditions to precisely convey the growth environment.

 In terms of formatting, consider placing the type strain first in line 17-18 and ensure consistent decimalization, such as in line 21 where "97.40%" is presented.

 Also, please provide details about the 16S rRNA gene sequence similarity and ANI values between the two KMM isolates in line 20.

Author Response

Responses for Reviewer #1: Thank you very much for reviewing our manuscript. We appreciate your valuable comments and suggestions. We have revised our manuscript thoroughly in accordance with your comments.

Comments: The authors skillfully argue for the introduction of the novel strains as a distinct genus based on the cpAAI criterion. This approach is in alignment with prior research indicating the limitations of 16S rRNA sequence-based phylogenetic analyses in precisely elucidating the lineage within a specific clade. Nevertheless, I suggest that for a clearer understanding of the relationship between the new strains and closely related species, a reanalysis of the 16S rRNA gene sequence-based phylogenetic tree (Figure 1) or a phylogenomic tree (Figure 2) is advisable. This reanalysis should include supplementary data containing a sufficient number of sequences from validly published Rhizobium species. Such an addition would substantially enhance the comprehension of the phylogenetic framework. Regarding Figure 1, it is recommended to consider alternate methods like Maximum Likelihood and Bayesian approaches, which incorporate sequence evolution models and are better suited for constructing phylogenies. Please show Maximum Likelihood tree and others in supplementary materials. Considering that the Molecular Evolutionary Genetics Analysis software (MEGA) has been updated to version 11.0.13, it is recommended to employ the latest version along with its corresponding references.

Response: According to your recommendation we have re-analyzed 16S rRNA phylogeny to clarify a taxonomic position of the new strains. We re-downloaded all 16S rRNA sequences with >95% similarity from the Ezbiocloud and LPSN servers (133 strains in total). The16S rRNA sequences of 50 strains from closely related genera (a limited number for the server) were then upload to the TYGS server. The phylogenetic analyses were repeated with an additional request to use ML and MP analyses. We have added the results (Lines 146-151, Figure 1). The reference to MEGA has been deleted because phylogenetic analyses was done on the TYGS server. New references under numbers 31-34 have been added. Based on the literature data, Bayesian approach as well as other ML and MP approaches do not help differentiate representatives of Rhizobiaceae members based on 16S rRNA gene.

Comment: A minor issue arises with the resolution of Figure 3, which currently stands at a level that impedes smooth interpretation.

Response: We are very sorry for the inconvenience caused. The problem is now solved.

Comment: An observation regarding line 271: "The core-genome shared by the four compared strains was represented by 2547 orthologous clusters (Figure 4a)." Might be 5 compared strains?

Response: Thank you very much. It was corrected (Line 276).

Comment: It would greatly enhance clarity if the symbols denoted by '(+)' within the parentheses in Table 2 were explicitly explained. At present, their meanings remain unspecified in the table legend.

Response: The symbols denoted by '(+)' was clarified (Line 339).

Comment: I recommend providing details about the cultivation conditions, as they could impact biosynthesis and subsequently alter chemotaxonomic characteristics like fatty acid profiles in bacteria. To ensure comparability, it is important to specify whether the experiments were conducted under the same conditions as the comparison strains. In this regard, please describe the incubation conditions and duration of cell incubation for the chemotaxonomic tests.

Response: It was corrected as: For polar lipid and fatty acid analyses, strains KMM 9553 and KMM 9576T and seven related type strains were cultivated on YMA at 28 °C for 48 h (Line 123).

Comment: Regarding line 381 and the term 'aerobic', I would appreciate clarification on whether the authors explored anaerobic growth conditions. It might be more accurate to denote 'strictly aerobic' or 'facultative anaerobic' growth conditions to precisely convey the growth environment.

Response: We used the term ‘aerobic' in the part of the genus description (Line 386) and we would like to leave it as is. No growth observed when they were tested for the anaerobic growth on MA 2216 for a week using an AnaeroPackTM (Mitsubishi Gas Chemical America, Inc.). The sentence was added to the manuscript text (Lines 86-88). In addition, strains KMM 9576T and KMM 9553 were negative for fermentation of D-glucose, arginine dihydrolase, lysine decarboxylase, ornithine decarboxylase, H2S production under anaerobic conditions and tryptophane deaminase, indole production and acetoin production (Voges-Proskauer reaction). All the characteristics are indicated in the species description.

Comment: In terms of formatting, consider placing the type strain first in line 17-18 and ensure consistent decimalization, such as in line 21 where "97.40%" is presented.

Response: It was corrected throughout the text as you recommended.

Comment: Also, please provide details about the 16S rRNA gene sequence similarity and ANI values between the two KMM isolates in line 20.

Response: It was corrected as you recommended (Lines 20, 21, 195, 213, 215).

Reviewer 2 Report

Romanenko et al. characterized two strains (KMM 17 9553 and KMM 9576T) isolated from a sandy sediment sample from the Sea of Japan seashore by using a polyphasic approach. The obtained results indicated that these two strains represent a novel species and a novel genus of the family Rhizobiaceae. However, in my opinion, this study should be further improved in order to be suitable for publication (see comments below).

Major comments

1.       The authors presented two core-proteome phylogenies. The first one (Fig. 2) was inferred using PhyloPhlAn software. However, it is unclear how was the second phylogeny (Fig. 3) inferred. The first phylogeny is missing some very important members of Rhizobiaceae family, such “R. album”. The second phylogeny is missing support values, in order to assess grouping of novel strains with “R. album”. The authors should use the same (extended) dataset to infer core-proteome phylogeny, including various members of Rhizobium genus (different clades), including “R. album”. Moreover, it is unlikely that the first phylogeny was based on the maximum number of proteins (400). Usually, some protein are missing from the dataset. Please indicate in the revised version the number of markers used for phylogeny.

2.       The introduction is confusing and should be thoroughly revised. It is unclear why the athors are mentioning the genus Blastobacter. It is true that one species was misclassified (Blastobacter capsulatus), and later transferred to the genus Pararhizobium. However, the genus Blastobacter is a bona fide genus from the family Nitrobacteraceae.

L54-55 cpAAI values were not used to prove monophyly. In the mentoned reference, the monophyly was demonstrated by phylogenetic analysis, while cpAAI values were used for genus delimitation (between monopyhletic groups).

L63-65: Rhizobia is generic term for a polyphyletic group of bacteria able to fix atmospheric nitrogen and nodulate leguminous plants. They can belong to genera Allorhizobium, Rhizobium, Sinorhizobium, Mesorhizobium (family Phyllobacteriaceae), Bradyrhizobium (family  Bradyrhizobiaceae) etc… In this respect, all rhizobia are nodulating. Please use this term properly. Other members of the family Rhizobiaceae can be plant pathogens and cause tumors on plants. Others are free-living and were isolated from plants, soil, water etc. They were also isolated from see sediments (see 10.1099/ijsem.0.003766 and 10.1007/s00284-023-03402-0), as strains characterized in this study, but it was not even mentioned in the introduction.

3.       Please include one or two sentences in which it is described that two strains analyzed are not of clonal origin, for instance that they differed as suggested by ANI, AAI, and dDDH.

Minor comments

L57: The name Xaviernesmea was validated, and should not be written under quotation marks.

L143-145: This fits better to the Results section.

L185: Why were two versions of the same software used?

L298-300: I would say that it was expected, as the strains were not isolated from nodules and a vast of Rhizobiaceae members do not possess nodulation genes.

L343: The name of the species was mentioned above, please abbreviate genus name, as M. mediterraneum. Please correct this for all other cases.

L388: Please indicate accession number of the genome sequence for the type strain in the protologue.

Moderate editing of English language required

Author Response

Responses for Reviewer #2: Thank you very much for reviewing our manuscript. We appreciate your valuable comments and suggestions. We have revised our manuscript thoroughly in accordance with your comments.

Major comments

Comment: The authors presented two core-proteome phylogenies. The first one (Fig. 2) was inferred using PhyloPhlAn software. However, it is unclear how was the second phylogeny (Fig. 3) inferred. The first phylogeny is missing some very important members of Rhizobiaceae family, such “R. album”. The second phylogeny is missing support values, in order to assess grouping of novel strains with “R. album”. The authors should use the same (extended) dataset to infer core-proteome phylogeny, including various members of Rhizobium genus (different clades), including “R. album”. Moreover, it is unlikely that the first phylogeny was based on the maximum number of proteins (400). Usually, some protein are missing from the dataset. Please indicate in the revised version the number of markers used for phylogeny.

Response: Thank you for your important comment. Indeed, some markers were discarded from the analysis due to their incompleteness in a part of genomes. The correct number of markers used for phylogeny reconstructed by PhyloPhlAn was 378. The second phylogeny (core-proteome phylogeny) are based on 170 (more correct was 169) non-recombining core protein markers conserved among a set of 97 Rhizobiaceae genomes as described by Kuzmanović et al. (2021). It was obtained using IQ-TREE software under the LG+F+I+I+R8 model with bootstrapping using 100 replicates. Support values not indicated for the tree in Fig. 3, because visualization of core-proteome phylogeny and cpAAI values between the strains and representatives of the family Rhizobiaceae was made as suggested in https://github.com/flass/cpAAI_Rhizobiaceae. We decided to remove the phylogeny reconstructed by PhyloPhlAn3 and prepare a figure containing the original tree for core-proteome phylogeny with support values. The corresponding information has been added to the Methods and Result Sections (Lines 174-177, 220). Please, see Figures 2 and 3.

Comment: The introduction is confusing and should be thoroughly revised. It is unclear why the athors are mentioning the genus Blastobacter. It is true that one species was misclassified (Blastobacter capsulatus), and later transferred to the genus Pararhizobium. However, the genus Blastobacter is a bona fide genus from the family Nitrobacteraceae.

Response: The sentence was corrected as: The use of MLSA demonstrated high heterogeneity of the genera Rhizobium and Agrobacterium within the family Rhizobiaceae, which led to reclassification of the species group “Rhizobium galegae” as a novel genus Neorhizobium [7]. In addition, based on the MLSA results, the genus named Pararhizobium was proposed to accommodate the cluster of Rhizobium giardinii, R. herbae, “R. helanshanense”, R. sphaerophysae, and Blastobacter capsulatus [7,8]. Please, find it in the Lines 46-51.

Comment: L54-55 cpAAI values were not used to prove monophyly. In the mentoned reference, the monophyly was demonstrated by phylogenetic analysis, while cpAAI values were used for genus delimitation (between monopyhletic groups).

Response: Thank you very much. It was corrected as “Recently, in the phylogenomic study based on whole genome sequencing analysis, Kuz-manovic et al. (2022) [9] have proposed a common approach for the genera delineation in the family Rhizobiaceae applying a pairwise core-proteome average amino acid identity (cpAAI) value of approximately 86%.”.

Comment: L63-65: Rhizobia is generic term for a polyphyletic group of bacteria able to fix atmospheric nitrogen and nodulate leguminous plants. They can belong to genera Allorhizobium, Rhizobium, Sinorhizobium, Mesorhizobium (family Phyllobacteriaceae), Bradyrhizobium (family Bradyrhizobiaceae) etc… In this respect, all rhizobia are nodulating. Please use this term properly. Other members of the family Rhizobiaceae can be plant pathogens and cause tumors on plants. Others are free-living and were isolated from plants, soil, water etc. They were also isolated from see sediments (see 10.1099/ijsem.0.003766 and 10.1007/s00284-023-03402-0), as strains characterized in this study, but it was not even mentioned in the introduction.

Response: We have already mentioned “non-symbiotic members of the family Rhizobiaceae recovered from aquatic and marine ecosystems….”. Also we have added two references as you recommended.

Cao J, Wei Y, Lai Q, Wu Y, Deng J, Li J, Liu R, Wang L, Fang J. Georhizobium profundi gen. nov., sp. nov., a piezotolerant bacterium isolated from a deep-sea sediment sample of the New Britain Trench. Int J Syst Evol Microbiol. 2020 Jan;70(1):373-379.

Wang XN, Wang L, He W, Yang Q, Zhang DF. Description of Flavimaribacter sediminis gen. nov., sp. nov., a New Member of the Family Rhizobiaceae Isolated from Marine Sediment. Curr Microbiol. 2023 Jul 26;80(9):301.

Comment: Please include one or two sentences in which it is described that two strains analyzed are not of clonal origin, for instance that they differed as suggested by ANI, AAI, and dDDH.

Response: The sentence was added as you recommended (Lines 213-215).

Minor comments

L57: The name Xaviernesmea was validated, and should not be written under quotation marks.

Response: It was corrected (Line 56).

L143-145: This fits better to the Results section.

Response: These sentences were transferred to the Results section (Lines 191-194).

L185: Why were two versions of the same software used?

Response: We started the analysis using an old version of Orthovenn. At the moment of finish, we were using a more modern version. We consider that both versions should be mentioned.

L298-300: I would say that it was expected, as the strains were not isolated from nodules and a vast of Rhizobiaceae members do not possess nodulation genes.

Response: We consider that these genes are key markers of the bacterial group. Therefore, they need to be analyzed and reported. We removed the words «striking feature» (Lines 352-353).

L343: The name of the species was mentioned above, please abbreviate genus name, as M. mediterraneum. Please correct this for all other cases.

Response: It was corrected throughout the text.

L388: Please indicate accession number of the genome sequence for the type strain in the protologue.

Response: The GenBank accession number for the whole-genome shotgun sequence of strain KMM 9576T was included to the protologue.

Moderate editing of English language was performed.

Round 2

Reviewer 2 Report

I would like to thank the authors for addressing all my comments. I have some minor suggestions.

 -          I suggest to reformulate the sentence in L54-55 as ““Recently, in the phylogenomic study based on whole genome sequencing analysis, Kuzmanovic et al. (2022) [9] have proposed a common approach for the genera delineation in the family Rhizobiaceae applying a core-genome gene phylogeny and a pairwise core-proteome average amino acid identity (cpAAI) threshold of approximately 86%.”.

 -          I suggest to reformulate the sentence in L213-215 to “Genome sequences of strains KMM 9576T and KMM 244 9553 exhibited   ANI/ AAI values of 99.5%/ 99.7%, and the dDDH values of 99.6%, confirming their non-clonal origin (Table S1).”

 -          L47-48: Please change to “in resolving taxonomic position of related species“

 -          Regarding the figure 3, it is not if the support values represent bootstrap values or the numbers on the nodes indicate the approximate Bayesian posterior probabilities support values as implemented in IQ-TREE?

Minor editing of English language required